# Overexpression of Neuregulin-1 Type III Has Impact on Visual Function in Mice

**DOI:** 10.3390/ijms23094489

**Published:** 2022-04-19

**Authors:** Nan Su, Weiqi Zhang, Nicole Eter, Peter Heiduschka, Mingyue Zhang

**Affiliations:** 1Research Laboratory, Department of Ophthalmology, University Hospital Muenster, 48149 Muenster, Germany; yanshengsu266@gmail.com (N.S.); nicole.eter@ukmuenster.de (N.E.); peter.heiduschka@ukmuenster.de (P.H.); 2Laboratory for Moleculare Neuroscience, Clinic for mental Health, University Hospital Muenster, 48149 Muenster, Germany; wzhang@uni-muenster.de

**Keywords:** electroretinography, visual evoked potential, neuregulin 1, schizophrenia

## Abstract

Schizophrenia is associated with several brain deficits, including abnormalities in visual processes. Neuregulin-1 (Nrg1) is a family of trophic factors containing an epidermal growth factor (EGF)-like domain. It is thought to play a role in neural development and has been linked to neuropsychiatric disorders. Abnormal Nrg1 expression has been observed in schizophrenia in clinical studies. Moreover, in schizophrenia, there is more and more evidence found about pathological changes of the retina regarding structural, neurochemical and physiological parameters. However, mechanisms of these changes are not well known. To investigate this, we analysed the function of the visual system using electroretinography (ERG) and the measurement of visual evoked potentials (VEP) in transgenic mice overexpressing Nrg1 type III of three different ages (12 weeks, 24 weeks and 55 weeks). ERG amplitudes tended to be higher in transgenic mice than in control mice in 12-week old mice, whereas the amplitudes were almost similar in older mice. VEP amplitudes were larger in transgenic mice at all ages, with significant differences at 12 and 55 weeks (*p* values between 0.003 and 0.036). Latencies in ERG and VEP measurements did not differ considerably between control mice and transgenic mice at any age. Our data show for the first time that overexpression of Nrg1 type III changed visual function in transgenic mice. Overall, this investigation of visual function in transgenic mice may be helpful to understand corresponding changes that occur in schizophrenia, as they may find use as biomarkers for psychiatric disorders as well as a potential tool for diagnosis in psychiatry.

## 1. Introduction

Schizophrenia is a complex disorder that affects 0.5–1% of the adult population throughout different ethnicities in the world. According to findings reported in the literature, several genes have been associated with the neuropathology in diverse populations [1]. Among these candidates, genes encoding the proteins neuregulin (Nrg1) and its receptor ErbB4 have been shown to be promising susceptibility genes of schizophrenia [1]. Nrg1 is a pleiotropic growth and differentiation factor, which can be classified in six major isoforms, (Nrg1 type I–VI). Types I, II, IV, V and VI are single transmembrane proteins, whereas type III contains a cysteine-rich domain that loops back intracellularly along with its N-terminal sequence [2]. Moreover, Nrg1 types1, II and III are best characterized in the peripheral nervous system [3]. Clinical studies show that some patients show abnormal levels of expression of Nrg1 and ErbB4 isoforms in different brain regions [4]. Schizophrenic patients are impaired in cognitive abilities, including executive control and working memory [5]. In addition, elevated levels of Nrg1 and ErbB4 proteins have been found in studies on post-mortem schizophrenic patients [6]. Furthermore, our early work on mice has shown that elevated Nrg1 expression (Nrg1-III-tg) showed ventricular enlargement and symptoms similar to schizophrenia [7]. Similarly, Olaya and colleagues showed in 2018 that overexpression of Nrg1 type III in mice confers schizophrenia-like behavior [8]. Importantly, another study confirmed that Nrg1/ErbB4 regulates visual cortical plasticity [9].

Researchers have long been aware of the link between schizophrenia and visual processing impairments, which are accompanied by multiple structural and functional disturbances in patients. Furthermore, the retina may be particularly affected, as it belongs to the central nervous system and shows similarities to the brain and spinal cord in terms of structure, functionality, response to insult, and immunology [10]. In addition, from an embryonic point of view, the retina and optic nerve, which have a neuroectodermal origin, emerge from diencephalon and can be seen with the naked eye in its natural state in the living organism [11]. Impairments of visual processing are well established in schizophrenia, including multiple structural and functional disturbances in patients. In addition, in studies in which patients were checked for factors such as psychotic symptoms and auditory distortions, visual distortions (including those of the retina) have been associated with suicidal ideation [12]. These alterations of the visual system include dopaminergic abnormalities, abnormal output, maculopathies and retinopathies, cataracts, poor visual acuity, and thinning of the retinal fibre layer (RNFL) [11]. Looking at this from another angle, some classic ocular pathologies have been found to occur in the context of several major neurodegenerative disorders, and RNFL thinning is related to brain volume loss in aging and illness progression as well as cognitive decline in multiple sclerosis and Alzheimer’s disease [13].

The purpose of this study was to check changes in the function of the visual system of transgenic mice overexpressing Nrg1 type III, as this protein is linked with both schizophrenia and visual processing. Visual function was assessed by the flash electroretinography (ERG) and measurement of visual evoked potentials (VEP). While several ERG anomalies have been identified in patients with psychiatric disorders [14], the underlying mechanisms and visual processing abnormalities in schizophrenia are still unknown. Moreover, we checked retinal morphology by optical coherence tomography (OCT). This method is non-invasive and rapid, providing the quantitative measurement of RNFL thickness and retinal volume (e.g., macula volume/MV) [15].

There are various proteins in the retina that could probably be influenced by overexpression of Nrg1 type III. We checked frozen sections of the eyes of control mice and transgenic mice for the immunoreactivity (IR) for an isoform of glutamic acid decarboxylase (GAD65), the voltage-gated K^+^ channel Kv2.1, and the postsynaptic density protein 95 (PSD-95).

## 2. Materials and Methods

### 2.1. Animals

The experiments were performed in accordance with European Communities Council Directive (86/EEC) and were approved by the Federal State Office for Consumer Protection and Food Safety of North Rhine-Westphalia (LANUV), Germany (file no. 84-02.04.2016.A417). All efforts were made to minimize animal suffering and to reduce the number of animals used in the experiments to the minimum necessary for reliable statistical analyses.

Animals of three different ages were investigated, young (12 weeks old), midlife (24 weeks old) and older (55 weeks old) mice. In each age group, transgenic and control mice were compared. Generation and genotyping of transgenic Nrg1-III-tg mice have been described in detail in [16].

### 2.2. Visual Electrophysiology

Electroretinography was performed as described previously in Schubert et al., 2015 [17]. Briefly, the mice were anaesthetised using a standard intraperitoneal ketamine/xylazine injection. Sleeping animals were placed on a heating pad to dilate the pupils by tropicamide and neosynephrine eye drops. Desensitisation of the cornea was achieved by a drop of proparacaine.

For the ERG and VEP measurement, the commercial measuring device RetiPort from Roland Consult (Brandenburg, Germany) was used. During the measurement, the animals were placed on a heated plate at 37 °C to prevent cooling of the animals. For the ERG measurement, a gold ring electrode was placed on the cornea of the left eye without damaging the cornea. VEP was recorded simultaneously by inserting a stainless-steel needle electrode subcutaneously above the visual centre of the mice on top of the skull between the ears. Another gold electrode that was moistened with saline and placed into the mouth of the animals served as the reference.

Measurement was performed in the scotopic mode, with animals that were dark-adapted for at least 12 h. Visual stimulation was performed by the application of flashes of six different light intensities, ranging from 0.0003 to 30 cd∙s/m^2^. Responses of the visual system were recorded, averaged, and stored by the RetiPort device. After the measurement, the still sleeping animals were kept in a separate box and given back into the cage after awakening.

### 2.3. OCT

To measure the thickness of the retina, in vivo imaging was performed. Mice were anaesthetised and the pupils dilated as described above. The mice were put in front of the “Spectralis” device by Heidelberg Engineering and were examined by optical coherence tomography (OCT). Formation of cataract was delayed by dropping distilled water onto the eyes.

Images were centered with the optic nerve head in the centre of the image. Retinal thickness was determined using the built-in routines in the nasal, superior, temporal and inferior quadrants of the inner circle and the outer circle of the ETDRS grid.

### 2.4. Immunohistochemistry

Eyes of euthanised mice were isolated and fixed in 4% paraformaldehyde for 1 h, washed 2× in PBS pH 7.4 for 5 min and frozen in NEG-50™. Cryo sections (thickness 10 µm) were cut using a Cryostar NX70 cryostat (Thermo Fisher Scientific, Waltham, MA, USA), placed on Starfrost Advanced Adhesive glass slides (Engelbrecht) and stored at −20 °C until used for immunohistochemistry.

For immunohistochemistry, sections were blocked with Power Block™ reagent (HK085-5K, BioGenex) at room temperature for 6 min, then washed 3× with 0.1 M PBS and incubated overnight with primary antibodies at 4 °C. The sections were then washed 3× with 0.1 M PBS and incubated with appropriate secondary antibodies for 1 h at room temperature. The nuclei were counterstained with DAPI (4′6′-diamidino-2-phenylindole dihydrochloride), diluted with pure water 1:300, for 7 min at room temperature. The primary antibody was diluted in 1% bovine serum albumin containing 0.1% Triton X-100, and the secondary antibody goat anti rabbit were diluted with 1% bovine serum albumin (Table 1). Finally, sections were washed 3× with 0.1M PBS and mounted under glass coverslips using mounting medium (ImmuMount™, Thermo Scientific).

### 2.5. Quantitative Analysis of Fluorescence in the Images of Histological Staining

Overview images of the mice retina sections were taken with 40× magnification (Axio Imager M2, Carl Zeiss, Jena, Germany). To ensure comparability of the multiple specimens, all procedures of staining and image acquisition were performed in an identical way. Digital images were processed using ImageJ (version 1.46r, NIH, Bethesda, MD, USA). Regions of interest (ROIs) were set manually in the ganglion cell, amacrine cell, and inner and outer plexiform layers using the implemented ROI manager, and the intensity of staining against KV, GAD, and PSD within the ROIs was measured to determine the amount of staining. To achieve reliable data, a uniform background in all areas of the image was necessary. The threshold tool settings were the same in every image.

### 2.6. Data Analysis

Data are presented as mean ± SEM. Evaluation of the data was performed by separately comparing the means of each parameter obtained in the control mice and the Nrg1-III-tg mice. After testing for normal distribution with a Shapiro-Wilk test, a parametric unpaired Student’s t-test or a non-parametric Mann–Whitney test wa used to determine the difference between control mice and Nrg1-III-tg mice. The level of statistical significance was set as *p* = 0.05, statistical significance is indicated as an * *p* < 0.05, ** *p* < 0.01, *** *p* < 0.001.

## 3. Results

### 3.1. Investigation of the Function of the Visual System Measured by ERG and VEP

ERG and VEP measurements were carried out in younger (12 weeks old), midlife (24 weeks old) and older (55 weeks old) mice; in these age groups, transgenic mice and control mice were compared.

In each age group, amplitudes of scotopic a-waves and b-waves displayed obvious differences. In younger mice, these amplitudes showed a trend to be smaller in control mice than in transgenic mice. However, amplitudes were higher in control mice than in transgenic mice in midlife mice and older mice, and the difference was statistically significant in midlife animals (Figure 1A). The latencies in scotopic ERG were in the same range in the three groups, and no relevant differences were seen (Figure 1B).

Amplitudes of scotopic oscillatory potentials were significantly higher in transgenic mice than in control mice in younger mice, however, not in midlife and older mice. No significant differences in the latencies of oscillatory potentials were found between the groups (Figure 1C). The b/a ratio tended to be slightly higher in control mice compared to transgenic mice (Figure 1D).

The amplitudes of scotopic VEP were significantly higher in the younger and older transgenic mice than in control mice at most light intensities (Figure 2A). Moreover, amplitudes of VEP were higher in transgenic mice than in control mice in the midlife mice, though without significance of the difference. Despite some deviations in the younger mice and the midlife mice, latencies did not show relevant differences between transgenic mice and control mice (Figure 2B).

Examples of waveforms of ERG and VEP measurements in young animals are shown in Figure 3. As a general observation, amplitudes of electrophysiological parameters decreased with increasing age. It appeared that amplitudes decreased more clearly in transgenic mice than in control mice.

### 3.2. Investigation of the Thickness and Volume of the Mouse Retina Measured by OCT

OCT measurements were carried out in the younger, midlife and elder mice, and thickness and volume of the retina were measured.

We did not find any significant differences in the thickness (data not shown) between transgenic mice and control mice at any age. Furthermore, there was no significant difference between transgenic mice and control mice when the volumes were compared (Figure 4).

### 3.3. Semi-Quantitative Analysis of Histological Staining

We first performed immunohistochemical staining against Nrg1. Immunoreactivity (IR) for Nrg1 was found in the ganglion cell layer, the inner nuclear layer, outer plexiform layer and in the photoreceptor inner segments (Figure 5). It was clearly higher in transgenic mice than in control mice in midlife animals. Conversely, the IR for Nrg1 appeared to be lower in transgenic mice than in control mice in the older animals (Figure 5).

To determine the effect of overexpression of Neuregulin on the GABAergic system in the retina, we checked retinal sections for GAD65 IR (Figure 6). GAD65 IR was seen in the inner plexiform layer, and it was higher in transgenic mice than in control mice in midlife animals. In older mice, no difference between the two groups was seen.

We observed changes of ERG amplitudes in transgenic mice compared to control mice (Figure 1). Because voltage-gated K^+^ channels play a role in the visual process, we checked IR for Kv2.1 in the retina. Kv2.1 IR was seen in the ganglion cell layer, in traces in the inner plexiform layer and amacrine cells, and particularly strongly in the photoreceptor inner segments. The intensity of Kv2.1 IR was higher in transgenic mice than in control mice in midlife animals. In older mice, the difference between the two groups was much weaker (Figure 6).

IR for PSD-95 was found on the outer rim of the outer plexiform layer, i.e., most probably in the region of the synapses of the photoreceptors. The intensity of PSD-95 IR was similar in both groups in midlife animals. It decreased in older animals, and this decrease appeared to be more obvious in transgenic mice than in control mice (Figure 7).

## 4. Discussion

In the current study, we investigated the influence of neuronal overexpression of Nrg1-Type III on the visual function in mice by electroretinography and VEP measurement. The above data demonstrate that ERG responses had a significantly reduced scotopic responses in transgenic mice compared to control mice in midlife and older, but not in the younger mice, where amplitudes were larger in transgenic mice. VEP amplitudes were enhanced in Nrg1 III-tg mice at all ages, with significance in younger and older mice. Latencies did not show any relevant differences between the groups at any age, either in ERG, or in VEP measurements.

The human neuregulin1 (Nrg1) gene is a major schizophrenia susceptibility gene, and its association with the illness has been found in different populations [18]. In addition, dysregulated expression of Nrg1, including elevation of expression of Nrg1, increases disease susceptibility and has been found in studies of post-mortem brain tissue from schizophrenia patients [19]. In our previous mouse studies, gene disruption that increased expression of Nrg1-Type III can confer distinct schizophrenia-like phenotypes in behaviour and brain biology [7].

More and more studies are showing that schizophrenia is associated with several brain deficits, including visual processing deficits [20,21]. As key player in visual processing, the retina is part of the central nervous system and is composed of several layers. To evaluate the function of specific layers of neurons of the retina, ERG can be used [22]. The cornea-negative a-wave indicates the electrical activity of the photoreceptors, and ON-bipolar cells are the source of the b-wave in the retina of mice [23,24]. Our results show that electroretinographic a-waves and b-waves were smaller in transgenic mice compared to control mice in the midlife and elder mice, but not in the younger mice, suggesting that Nrg1-III-tg mice had impaired retinal function at higher ages. Considering the origin of the a-waves and b-waves and the transmission and processing of nerve signals in the retina as well as to better understand the function and mechanism of Nrg1 in the retina, we then performed a series of immunostaining. As has been found in the brain, the Nrg1 signalling pathway has an effect on neurotransmission and synaptic plasticity [7]. Hence, further studies are required to determine which changes in the neural signalling occur in the retina of transgenic mice. Nrg1 has been implicated in neural excitatory synapse development and regulated neuronal migration in the neuronal system [25,26]. The inhibitory neurotransmitter GABA is synthesized by two isoforms of glutamic acid decarboxylase (GAD65, GAD67) [27], and mice lacking GAD 65 show an impaired visual cortical plasticity [28]. The 65-KD isoform (GAD65) is found primarily in the axon terminals in the visual cortex and reversibly bound to the membrane of synaptic vesicles, playing a role in the control of the synaptic release of GABA [29,30]. We found an enhanced IR of GAD65 in midlife transgenic mice, but not in older mice, implying increased GABA synthase and release from the synaptic vesicle in retina. On the other hand, we have previously shown that electrophysiological recordings in an acute cortical slice revealed a moderate increase in frequency [7]. Whether neuregulin has a common mechanism of affecting synaptic transmission and synaptic plasticity in the retina and the cortex, requires further studies. The amount of neurotransmitter release depends on the neuronal membrane excitability, shape of the action potential, or probability of vesicular release [31,32]. In addition, potassium channels regulate neuronal excitability, setting resting membrane potentials and decreasing and increasing excitability [33]. The voltage-gated Kv2.1subunit is nearly ubiquitously expressed in the mammalian brain and is present in most neuronal cell types, including pyramidal cells and interneurons and plays an important role in its ability to dynamically contribute to neuronal excitation [34,35]. The results show that overexpression of NRG1 has an impact on IR of KV2.1 both in the amacrine cell layer and in the photoreceptor inner segments layer in midlife mice. Meanwhile, these data are consistent with ERG data that overexpression of Nrg1 affects the a-wave mainly in midlife mice, as the a-wave is generated mainly by the rod and cone receptor photocurrent [36]. However, there were no differences in older mice (Figure 6), suggesting an interference of development of Kv2.1 in the overexpression of Nrg1transgenic mice. Moreover, Nrg1 regulates the stability of PSD-95 in GABAergic neurons in a way that requires its receptor ErbB4 [37]. Reduced PSD-95 IR in older mice suggest that NRG1 regulates the PSD-95 in the synaptic terminal in the retina of transgenic mice.

The visual evoked potential (VEP) is an electrical potential recorded from the visual cortex in response to a visual stimulus. Therefore, the function of the optic nerve and the quality of signal processing in the visual cortex can be evaluated by measuring the VEP [38]. We observed an increase in the VEP amplitudes in transgenic mice. It may be caused by the presence of abnormal ganglion cells or an abnormal number of ganglion cells. A changed signal processing in the visual cortex is also possible. This must be evaluated in further studies.

We observed that there was more IR for Nrg1 in the retina of young transgenic mice than in control mice (Figure 5). As overexpression of Nrg1 is performed under the control of the Thy-1 promoter, it could be anticipated that mainly retinal ganglion cells showed immunoreactivity for Nrg1, because these cells especially produce Thy-1 at young ages [39]. Given that the expression of Nrg1 seems to change with age, this may suggest that NRG/ErbB signalling plays a role during the early development of the retina.

Our previous work showed that dysregulation of Nrg1 by cortical pyramidal neurons disrupts GABAergic and glutamatergic neurotransmission in cortex as well as synaptic plasticity [7]. All these changes in transgenic mice have the potential to alter the visual system and to further eventually impact ERG and VEP.

Given the findings of the current study, further investigations should more deeply explore the mechanisms by which these visual anomalies occur, as they will be helpful for understanding the biological basis of the psychiatric disorder. Taken together, these results indicate a critical role of Nrg1 on retinal synaptic system.

## Figures and Tables

**Figure 1 ijms-23-04489-f001:**
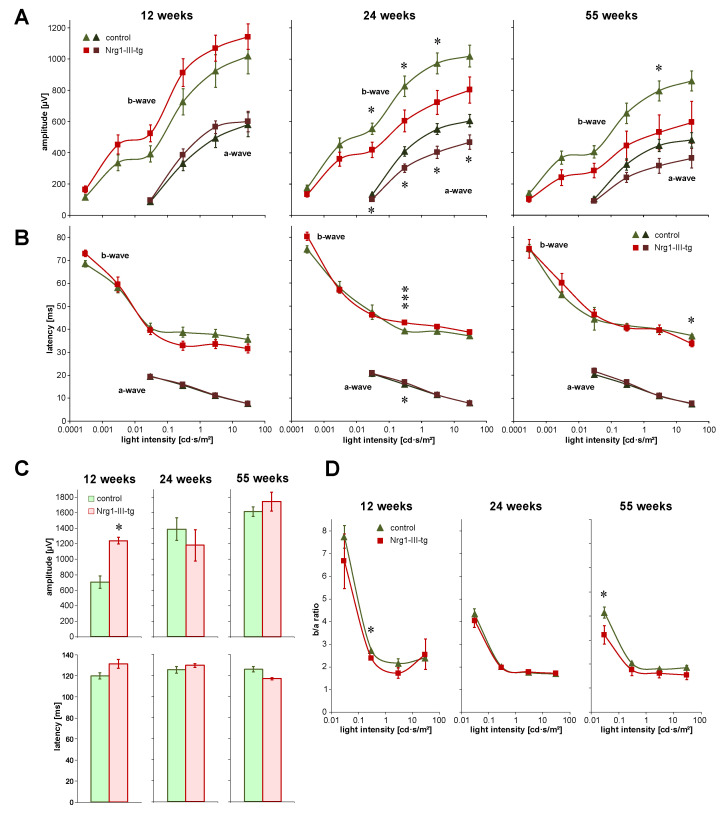
ERG Parameters in 12-, 24 and 55-week control and transgenic mice. Scotopic amplitudes (**A**) and latencies (**B**) of a-waves and b-waves at different light intensities as indicated: (**C**): amplitudes and latencies of scotopic oscillatory potentials; (**D**): the ratio of amplitudes of b-waves and a-waves. Number of animals at the age of 12, 24 and 55 weeks: *n* = 8, *n* = 13 and *n* = 9 control mice and *n* = 6, *n* = 5 and *n* = 8 Nrg1-III-tg age-matched mice, respectively. Data are shown as mean values. Error bars show standard error of means. Statistical significance of differences is indicated as follows: * *p* < 0.05, *** *p* < 0.001.

**Figure 2 ijms-23-04489-f002:**
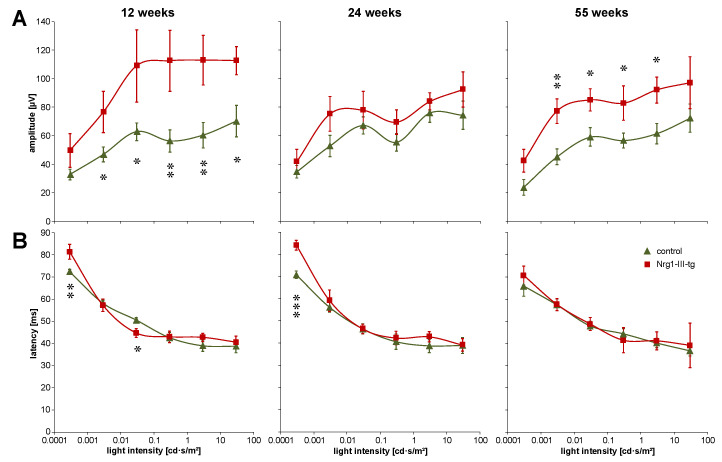
Scotopic VEP amplitudes and latencies in 12-, 24- and 55-week-old mice. Scotopic amplitudes (**A**) and latencies (**B**) of visual evoked potentials at different light intensities as indicated. Data are shown as mean values. Error bars show standard error of means. Number of animals at the age of 12, 24 and 55 weeks: *n* = 8, *n* = 13 and *n* = 9 control mice and *n* = 6, *n* = 5 and *n* = 8 Nrg1-III-tg age-matched mice, respectively. Statistical significance of differences is indicated as follows: * *p* < 0.05, ** *p* < 0.01, *** *p* < 0.001.

**Figure 3 ijms-23-04489-f003:**
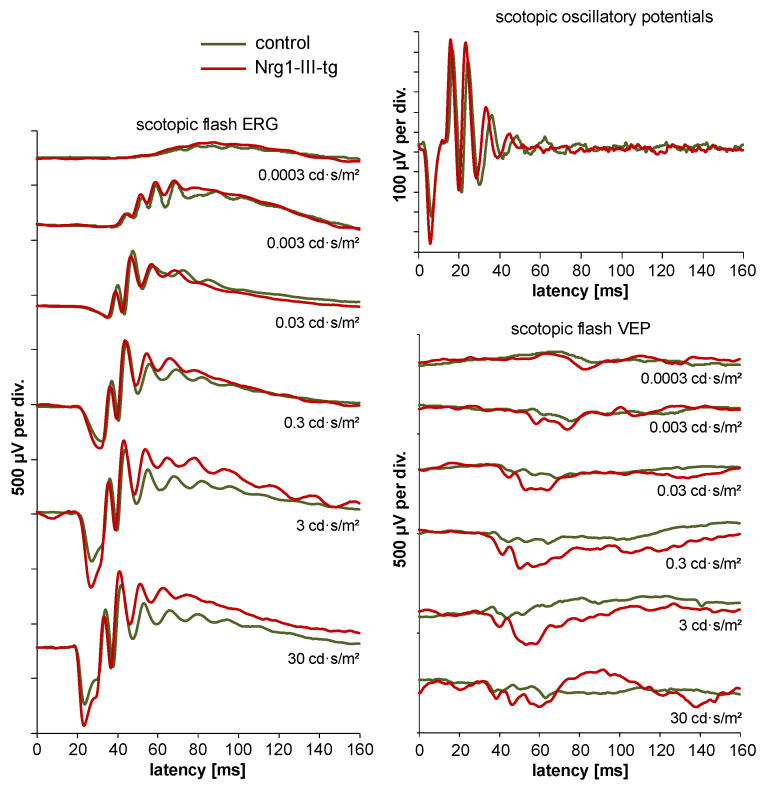
Typical waveforms of ERG and VEP measurements in 12-week-old control and transgenic mice.

**Figure 4 ijms-23-04489-f004:**
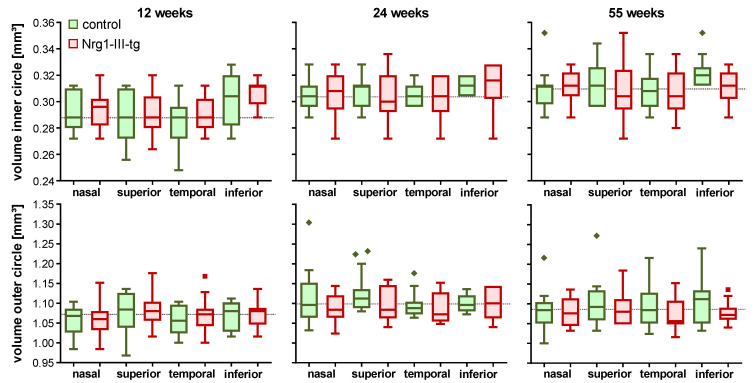
The Volume of retina in 12-, 24- and 55-week-old mice. Data are shown as mean values. Error bars show standard error of means. Number of animals at the age of 12, 24 and 55 weeks: *n* = 12, *n* = 13 and *n* = 14 control mice and *n* = 16, *n* = 8 and *n* = 10 Nrg1-III-tg age-matched mice, respectively.

**Figure 5 ijms-23-04489-f005:**
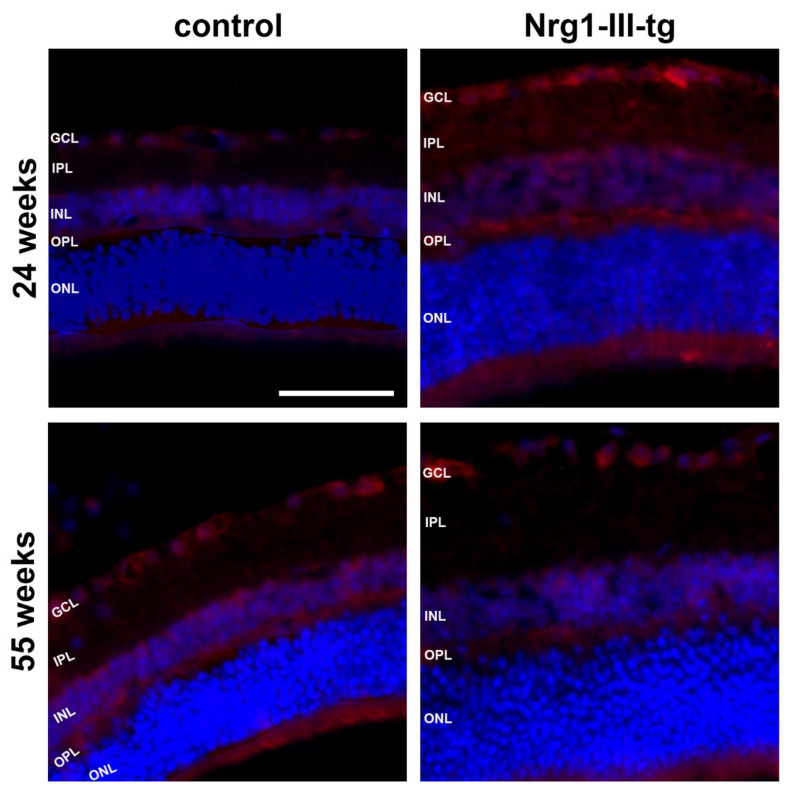
Immunohistochemical staining against neuregulin. The trophic factor neuregulin (red) in frozen sections of the retina of the mouse eye at 24 weeks and 55 weeks of control and Nrg1-III-tg as indicated. The cell nuclei were stained with DAPI (blue); GCL-ganglion cell layer; IPL-inner plexiform layer; INL-inner nuclear layer; OPL-outer plexiform layer; ONL-outer nuclear layer. Scale bar: 50 μm.

**Figure 6 ijms-23-04489-f006:**
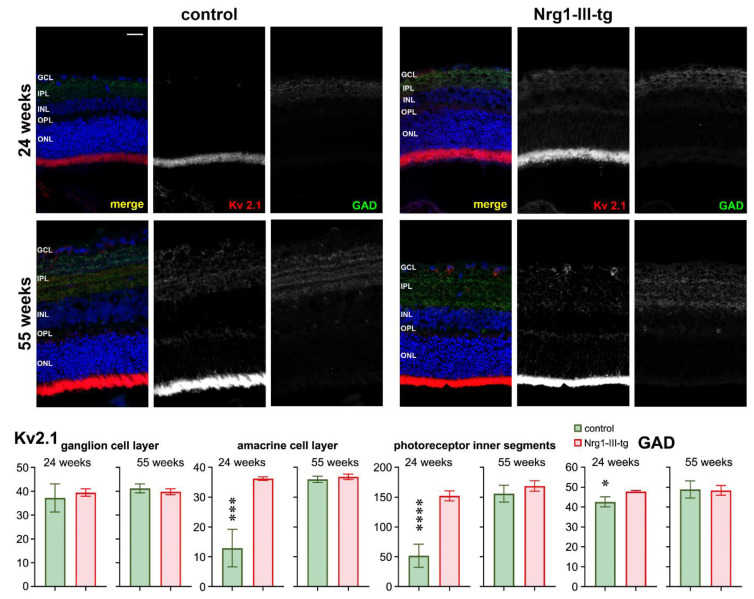
Immunohistochemical staining against Kv2.1 and GAD. Kv2.1 (red) and GAD (green) in frozen sections of the retina of the mouse eye at 24 weeks and 55 weeks of control and Nrg1-III-tg as indicated. The cell nuclei were stained with DAPI (blue); GCL-ganglion cell layer; IPL-inner plexiform layer; INL-inner nuclear layer; OPL-outer plexiform layer; ONL-outer nuclear layer. Scale bar: 50 μm. * *p* < 0.05, *** *p* < 0.001, **** *p* < 0.0001.

**Figure 7 ijms-23-04489-f007:**
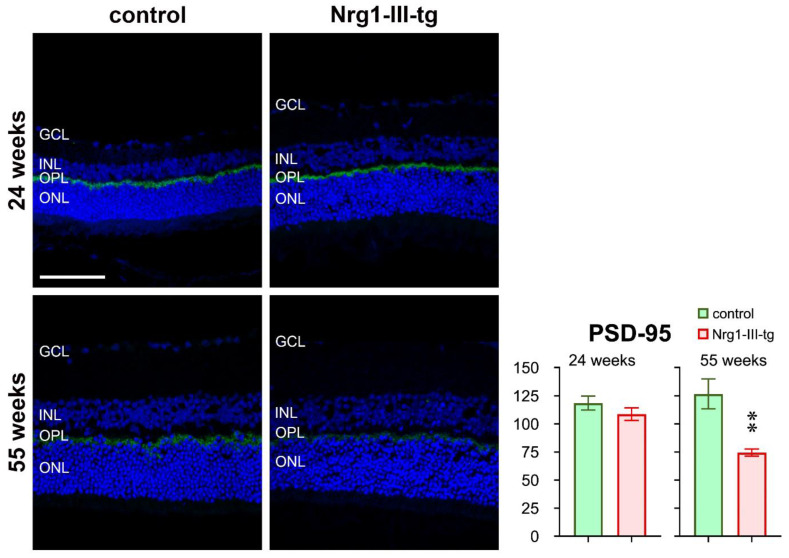
Immunohistochemical staining against PSD-95. PSD-95 (green) in frozen sections of the retina of the mouse eye at 24 weeks and 55 weeks of control and Nrg1-III-tg as indicated. The cell nuclei were stained with DAPI (blue); GCL-ganglion cell layer; IPL-inner plexiform layer; INL-inner nuclear layer; OPL-outer plexiform layer; ONL-outer nuclear layer. Scale bar: 50 μm. ** *p* < 0.01.

**Table 1 ijms-23-04489-t001:** Antibodies used in this study.

Primary Antibodies					
Specificity	Host		Supplier	Catalogue No.	Dilution
Kv-2.1	Rabbit		Synaptic Systems	231 002	1:500
GAD 2/GAD 65	Guinea pig		Synaptic Systems	198104	1:500
Neuregulin-1	Rabbit		GeneTex	GTX133355	1:1000
PSD-95	Rat		Santa Cruz Biotechnology	sc-32290	1:300
**Secondary Antibodies**					
**Specificity**	**Host**	**dye**	**Supplier**	**Catalogue No.**	**Dilution**
Anti-Rabbit	goat	Texas Red	Abcam	ab150080	1:800
Anti-Guinea pig	goat	Alexa Fluor 488	Abcam	ab150185	1:600

## Data Availability

Not applicable.

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
