# Peer review of "Overexpression of Neuregulin-1 Type III Has Impact on Visual Function in Mice"

_ijms, 2022, doi:10.3390/ijms23094489_

Round 1

Reviewer 1 Report

The manuscript by Su and colleagues addresses a practically significant question of phenotypic manifestation in response to overexpression of the probable determinant of schizophrenia, the Nrg1 signaling molecule. The mouse model of the disease is a strong tool in efforts to describe the etiology and develop therapeutic approaches. The authors applied a comprehensive approach to the lifetime and preparative study of the effect of Nrg1 misexpression on visual function. The quantity and quality of the described results do not allow us to doubt the importance of the publication of this work. The authors adequately interpret the presented data. These new results are important for the progress of fundamental biomedicine and molecular biology.

Author Response

The manuscript by Su and colleagues addresses a practically significant question of phenotypic manifestation in response to overexpression of the probable determinant of schizophrenia, the Nrg1 signaling molecule. The mouse model of the disease is a strong tool in efforts to describe the etiology and develop therapeutic approaches. The authors applied a comprehensive approach to the lifetime and preparative study of the effect of Nrg1 misexpression on visual function. The quantity and quality of the described results do not allow us to doubt the importance of the publication of this work. The authors adequately interpret the presented data. These new results are important for the progress of fundamental biomedicine and molecular biology.

Response:

We are appreciate your time and comments. As suggested by reviewer 1, we have revised and added some additional information in the abstract (Line 22) and introduction section in the line 36-39. There are also some minor English errors that have been corrected.

Reviewer 2 Report

The authors present their data regarding the function of the visual system using electroretinography (ERG) and measurement of visual evoked potentials (VEP) in a transgenic mouse overexpressing Nrg1 type III, possibly linked to schizophrenia.

This is a well written manuscript with a straightforward methodology, clear tables and figures. Therefore, I believe this manuscript can be accepted after some adjustments plus an elaboration on several topics as highlighted below:

  • Abstract: “Abnormal Nrg1 expression has been observed in schizophrenia”; please state if this is the case for preclinical and/or clinical studies?
  • abstract: “with significant differences in 12-weeks and 55-weeks old mice.”: please indicate p-values
  • abstract + introduction: “overexpressing Nrg1 type III” => how about the other types? please elaborate briefly about type I, II and IV, V and VI??
  • general: is there a relation between Nrg1 and dopamine?
  • methods: “Parametric unpaired Student’s t-test was used to determine difference between con-148 trol mice and Nrg1-III-tg mice.” => => do you know if your data are normally distributed? did you perform any normality tests? Since a Mann-Whitney test would be more accurate if the data did not pass this normality test (e.g. a D’Agostino and Pearson omnibus normality test)? Please attach these data to the manuscript.
  • methods: “isoform of glutamic acid decarboxylase 74 (GAD65), the voltage-gated K+ channel Kv2.1, and the postsynaptic density protein 9575 (PSD-95)” => why these specific proteins? can you refer to other studies? selection bias? Would it be interesting to analyze a protein involved in the dopaminergic pathways?
  • small remarks: typo in legend figure 4: “mice, respectively. .”; typo in discussion: “und mice lacking” => “and”; “However, there were no difference in”=> differences
  • Discussion: I am missing a limitation section? moreover, schizophrenia could be studied by behavioral analyses e.g. an open field test or building a nest or life span or…? These interesting analyses are missing in this study so please provide them in the limitation section
  • discussion: could this be a model for drug discovery? Would it be possible to develop a drug such as an Nrg1 type III-“antagonist”? please briefly elaborate
  • discussion: relationship Nrg1 type III: is it only schizophrenia or also other diseases, eg Charcot–Marie–Tooth type; please elaborate

Author Response

We appreciated the reviewer’s Comment. Please see the attachment for the revision manuscript we have changed. Thank you very much for time and suggestions! 

The followings are point by point response:

Point1: Abstract: “Abnormal Nrg1 expression has been observed in schizophrenia”; please state if this is the case for preclinical and/or clinical studies?

Response: Many thanks for your comment. This is the case for clinical studies. We revised the lines 14 and 15 in the abstract.

Point2: abstract: “with significant differences in 12-weeks and 55-weeks old mice.”: please indicate p-values

Response: Thank you very much for your comments. We added a range of p values in line 22.

Point3: abstract + introduction: “overexpressing Nrg1 type III” => how about the other types? please elaborate briefly about type I, II and IV, V and VI??

Response: Thank you very much for your advice. We included a briefly description in the lines 35-40.

Point4: general: is there a relation between Nrg1 and dopamine?

Response: Thank you very much for the question. In fact, there is a relation between Nrg1 and dopamine. In a 2012 article, Buonnano and his colleagues investigated the relationship between NRG/Erb and dopamine signalling in hippocampal gamma oscillations in rat. They reported that D4 receptor activation is essential for the effect of Nrg-1 on the network activity (see reference 1).

Point5.methods: “Parametric unpaired Student’s t-test was used to determine difference between con-148 trol mice and Nrg1-III-tg mice.” => => do you know if your data are normally distributed? did you perform any normality tests? Since a Mann-Whitney test would be more accurate if the data did not pass this normality test (e.g. a D’Agostino and Pearson omnibus normality test)? Please attach these data to the manuscript.

Response:

Many thanks for your comment. After having collected the data, first we checked whether the data were normally distributed using Shapiro-Wilk test or D’ Agostino& Pearson test. To analyse data that are not normally distributed, we usually use Mann-Whitney test compare them. In case of normally distributed data, we perform the Student’s t-test, as it has been the case in our manuscript.

To avoid ambiguity, we include explanations om the data analysis in the Line 152-154

Point6: methods: “isoform of glutamic acid decarboxylase 74 (GAD65), the voltage-gated K+ channel Kv2.1, and the postsynaptic density protein 9575 (PSD-95)” => why these specific proteins?

can you refer to other studies? selection bias? Would it be interesting to analyze a protein involved in the dopaminergic pathways?

Response: many papers have mentioned and demonstrated that Nrg/Erb signaling has a function in neurotransmission and neuroplasticity. In our previous paper (Agarwal et al 2014), we also showed that dysregulated expression of Neuregulin-1 causes unbalanced excitatory-inhibitory neurotransmission and altered synaptic plasticity.  In this manuscript, we observed changes in the electroretinographic responses and visual evoked potentials in transgenic mice. The changes in ERG are most likely due to changes in neurotransmitters in the retina, while synaptic transmission changes reflected in changed VEP are also involved in the pathogenesis of schizophrenia. One of the important neurotransmitters is glutamate. The glutamatergic theory of schizophrenia presupposes the hypofunctional of NMDA receptor, and, in the case of the retina, can potentially lead to the destruction of ganglion cells (DC Golf and JT Coyle 2001). NMDA receptors are principal regulators of synaptic signaling in the brain. Modulation of NMDA receptors function and trafficking is important for the regulation of synaptic transmission and synaptic plasticity. PSD-95 is a pivotal postsynaptic scaffolding protein in excitatory neurons and stabilizes NMDA receptors. Meanwhile, it is also a frequently used as a post synaptic marker protein for glutamatergic neurons.

Potassium channel are important regulators of cellular excitability, functioning to modulate the amplitude, duration and frequency of action potentials and firing patterns in the retina, and are also important in retinal ganglion cells development and protection. Loss of the KV2.1 greatly can cause ionic dysregulation and degeneration in rod photoreceptors (Zhong YS et al 2013.,; C Fortenbach 2021). All these changes may lead to abnormal ERG and VEP.

Another neurotransmitter in the mammalian retina is gamma-aminobutyric acid (GABA). Numerous data show their potential role in the pathogenesis of schizophrenia, however, the impact of GABA in retina is less known. In the retina, inhibitory process play a role mainly in horizonal and bipolar cells. Since there are several types of interneurons that may act in the retina, we usually first consider the most common GABAegic neuronal marker (GAD65), which is a specific marker for GABAergic neurons, without looking at a specific type of interneuron marker such as Parvalbumin interneuron (PV interneuron).

Dopaminergic synaptic transmission and pathway has to be checked, too. It is mainly produced by amacrine and interplexiform cells and shows diffusion through individual retinal layers. Thank you very much for your suggestion, and we will consider and added in the next article of our follow up study.

Point7 small remarks: typo in legend figure 4: “mice, respectively. .”; typo in discussion: “und mice lacking” => “and”; “However, there were no difference in”=> differences

Response: Thank you for pointing out these mistakes, we corrected them in lines 208, 278 and 298.

Point8: Discussion: I am missing a limitation section? moreover, schizophrenia could be studied by behavioral analyses e.g. an open field test or building a nest or life span or…? These interesting analyses are missing in this study so please provide them in the limitation section

Response: Thank you very much for your comments. In our previous paper 2014 (Agarwal et al 2014), a series of behavioral experiments has been conducted in tg mice such as open field, cued and contextual fear conditioning, MK-801 treatment and prepulse inhibition etc. 

Point9: discussion: could this be a model for drug discovery? Would it be possible to develop a drug such as an Nrg1 type III-“antagonist”? please briefly elaborate

Response: Thank you very much for your suggestions. Actually, the Munich Psychiatric Hospital is working on a related project, and they have published a paper based on the animal research (MC Wehr et al 2017). At the moment, they shall be working on the clinical study.

Point10: discussion: relationship Nrg1 type III: is it only schizophrenia or also other diseases, eg Charcot–Marie–Tooth type; please elaborate

Response: Many thanks for your comments. It is not only schizophrenia, but NRG/Erbb signalling mechanisms are also involved in other psychiatric disorder like bipolar or depression. In addition to that in the PNS, axonal Nrg1 is a key signaling protein that regulates the myelinating Schwann cells. As you mentioned, CMT disease is due to axonal loss and thereby leads to muscle atrophy accompanied with weakness. We do not know much about mechanisms between the CMT and Neuregulin. An experimental group at the MPI in Göttingen performed a lot of related studies. In an article they published in Nature Medicine, they proposed that  early neuregulin-1 treatment promotes Schwann cell differentiation, preserves axons and restores nerve function in rats’ disease model (Fledrich et al 2014)

[1]         R. H. Andersson et al., “Neuregulin and dopamine modulation of hippocampal gamma oscillations is dependent on dopamine D4 receptors,” Proc. Natl. Acad. Sci. U. S. A., vol. 109, no. 32, pp. 13118–13123, Aug. 2012, doi: 10.1073/PNAS.1201011109/SUPPL_FILE/PNAS.201201011SI.PDF.

[2]         A. Agarwal et al., “Dysregulated expression of neuregulin-1 by cortical pyramidal neurons disrupts synaptic plasticity,” Cell Rep., vol. 8, no. 4, pp. 1130–1145, Aug. 2014, doi: 10.1016/j.celrep.2014.07.026.

[3]         D. C. Goff and J. T. Coyle, “The Emerging Role of Glutamate in the Pathophysiology and Treatment of Schizophrenia,” 2001.

[4]         C. Fortenbach et al., “Loss of the K+ channel Kv2.1 greatly reduces outward dark current and causes ionic dysregulation and degeneration in rod photoreceptors,” J. Gen. Physiol., vol. 153, no. 2, Jan. 2021, doi: 10.1085/JGP.202012687/211728.

[5]         Y. S. Zhong, J. Wang, W. M. Liu, and Y. H. Zhu, “Potassium ion channels in retinal ganglion cells (review),” Mol. Med. Rep., vol. 8, no. 2, pp. 311–319, Aug. 2013, doi: 10.3892/MMR.2013.1508.

[6]         M. C. Wehr et al., “Spironolactone is an antagonist of NRG1-ERBB4 signaling and schizophrenia-relevant endophenotypes in mice,” EMBO Mol. Med., vol. 9, no. 10, pp. 1448–1462, Oct. 2017, doi: 10.15252/EMMM.201707691.

[7]         R. Fledrich et al., “Soluble neuregulin-1 modulates disease pathogenesis in rodent models of Charcot-Marie-Tooth disease 1A,” Nat. Med., vol. 20, no. 9, pp. 1055–1061, Sep. 2014, doi: 10.1038/NM.3664.

Reviewer 3 Report

The authors analysed function of the visual system using electroretinography and measured visual evoked potentials in a transgenic mouse overexpressing Nrg1 type III in mice at different ages. They showed that overexpression of Nrg1 type III changed visual function in transgenic mice. The results, as the authors suggest,  may help to understand corresponding changes that occur in schizophrenia, as they may find use as markers for psychiatric disorders as well as a potential tool for diagnosis in psychiatry.
Essentially I have no criticism regarding this manuscript as it was meaningfully planned, executed and written.
Please adapt the citation style to the editorial requirements.

Author Response

The authors analysed function of the visual system using electroretinography and measured visual evoked potentials in a transgenic mouse overexpressing Nrg1 type III in mice at different ages. They showed that overexpression of Nrg1 type III changed visual function in transgenic mice. The results, as the authors suggest,  may help to understand corresponding changes that occur in schizophrenia, as they may find use as markers for psychiatric disorders as well as a potential tool for diagnosis in psychiatry.
Essentially I have no criticism regarding this manuscript as it was meaningfully planned, executed and written.
Please adapt the citation style to the editorial requirements.

Response: Thank you very much for your time and comments. As suggested by reviewer 1, we made some minor additions and changes, please see in the manuscript in the attachment. Thanks again for your time and comments.

This manuscript is a resubmission of an earlier submission. The following is a list of the peer review reports and author responses from that submission.